# A systematic review and meta-analysis of victimisation and mental health prevalence among LGBTQ+ young people with experiences of self-harm and suicide

**A. Jess Williams** [1,2]*, **Christopher Jones**[3], **Jon Arcelus**[4], **Ellen Townsend**[2], **Aikaterini Lazaridou**[1], **Maria Michail**[1]

**1** School of Psychology, Institute for Mental Health, University of Birmingham, Birmingham, United Kingdom, **2** School of Psychology, Self-Harm Research Group, University of Nottingham, Nottingham, United Kingdom, **3** School of Psychology, Centre for Applied Psychology, University of Birmingham, Birmingham, United Kingdom, **4** School of Medicine, Institute of Mental Health, University of Nottingham, Nottingham, United Kingdom

* a.williams.10@pgr.bham.ac.uk

**Data Availability Statement:** All files are available from the Open Science Framework database: https://osf.io/2npgz/.

## Abstract

### Background

LGBTQ+ youth have higher rates of self-harm and suicide than cisgender, heterosexual peers. Less is known about prevalence of risks within these populations.

### Objectives

The first systematic review and meta-analysis to investigate the prevalence of risks among young people throughout the LGBTQ+ umbrella with experiences across the dimension of self-harm, suicidal ideation and suicide behaviour; and how they may differ between LGBTQ+ umbrella groups.

### Data sources

MEDLINE, Scopus, EMBASE, PsycINFO, and Web of Science searches were run to identify quantitative research papers (database inception to 31st January, 2020).

### Study eligibility criteria

Articles included were empirical quantitative studies, which examined risks associated with self-harm, suicidal ideation or suicidal behaviour in LGBTQ+ young people (12–25 years).

### Synthesis methods

2457 articles were identified for screening which was completed by two independent reviewers. 104 studies met inclusion criteria of which 40 had data which could be meta-analysed in a meaningful way. This analysis represents victimisation and mental health difficulties as risks among LGBTQ+ youth with self-harm and suicide experiences. Random-effects modelling was used for the main analyses with planned subgroup analyses.

**Funding:** This project was funded as part of an Economic and Social Research Council grant on the Doctoral Training Pathway. The lead author, A. Jess Williams, receives a student stipend from the ESRC. The funders had no role in study design, data collection and analysis, decision to publish, or preparation of the manuscript.

**Competing interests:** No - The authors have declared that no competing interests exist.

## Results

Victimisation and mental health were key risk factors across the dimension self-harm and suicide identified through all analyses. A pooled prevalence of 0.36 was indicated for victimisation and 0.39 for mental health difficulties within LGBTQ+ young people with experiences of self-harm or suicide. Odds ratios were calculated which demonstrated particularly high levels of victimisation (3.74) and mental health difficulties (2.67) when compared to cisgender, heterosexual counterparts who also had these experiences.

## Conclusions

Victimisation and mental health difficulties are highly prevalent among LGBTQ+ youth with experiences of self-harm and suicide. Due to inconsistency of reporting, further risk synthesis is limited. Given the global inclusion of studies, these results can be considered across countries and inform policy and suicide prevention initiatives.

## PROSPERO registration number

CRD42019130037.

## Introduction

Worldwide, suicide is one of the leading causes of death for young people [1], with adolescent suicide rates between 11.2–12.7 per 100,000 across low-, middle-, and high-income countries [2]. Suicidal thoughts and attempt are thought to be around 3 times higher among sexual orientation minorities (Lesbian, gay, bisexual, questioning or queer, LGBQ) youth when compared to heterosexual, cisgender counterparts [3]. A recent meta-analysis found suicidal ideation prevalence was demonstrated to be around 28% among gender identity minority groups (transgender and gender non-conforming, TGNC) and suicidal attempt prevalence was 14.8% [4]. Self-harm (defined as self-injury or self-poisoning of self, irrespective of suicidal intent [5]) is known as the most influential risk factor for completed suicide among young people [6,7]. There is also strong evidence that demonstrates the high prevalence of self-harm among young people who identify as LGBTQ+ (Lesbian, Gay, Bisexual, Transgender, Queer or Questioning, and others) [8]. Within LGBQ youth self-harm was reported by 65% of the sample whilst around 46% of TGNC samples have also reported this type of behaviour [9,10].

Among young people generally, regardless of sexual orientation or gender identity, risks associated with experiences of self-harm and suicide are numerous, ranging from childhood neglect to poor academic performance [11,12]. Given this, risk factors are often put into broad categories; demographic, psychosocial, mental health, or psychopathology etc. [13–15]. Within a category such as demographic risks, the individual risk factor can also range widely e.g. age [16], race [17,18] or education level [19]. Additionally, certain populations may also experience risks which are only influential to that specific group of individuals. LGBTQ+ young people are often exposed to additional stressors which are specifically related to their sexual orientation and gender identity when compared to cisgender heterosexual peers, such as institutionalised prejudice, social pressure and victimisation [20–22]. Among the LGBTQ+ umbrella there is also variation of how prevalent a risk may be to a subgroup. For example, someone who is outwardly gender nonconforming may receive more harassments than a cisgender member of the LGBTQ+ umbrella. Therefore, it is possible that there is another layer of risks which TGNC young people face. Gender nonconformity, gender dysphoria, and frustrations due to the long

waiting lists for gender affirming medical interventions are common among TGNC populations and have previously been shown to influence suicidal behaviour [23]. Although we know that negative experiences such as institutional prejudice, social pressures, victimisation are associated with self-harm or suicide among those who identify as LGBTQ+ young people [20–22,24], less is known about how prevalent these experiences may be within this population. This systematic review seeks to comprehensively investigate the prevalence of all risks within LGBTQ + young people who have a history of self-harm, suicidal ideation or attempt. Previous reviews in this population specifically focus on a category of self-harm and suicide; either non-suicidal self-injury or suicide excluding self-harm [25,26]. However, we aim to investigate outcomes across the dimension of self-harm, irrespective of intent, suicidal ideation and attempt to consider differences and similarities within risk prevalence by outcome among LGBTQ+ young people. This will allow us to explore risks across the dimensional structure of self-destructive thoughts and behaviours [27] and consider the comparison of risk across the continuum of suicidal intent. Furthermore, previous reviews have not looked at the prevalence of risk factors for self-harm and suicide across the full LGBTQ+ umbrella, therefore, losing comparability of risks within this broad population [28]. In this study, we consider LGBTQ+ young people as a whole group, and then by sexual orientation minority and gender identity minority groups.

## Objectives

1. To investigate, for the first time, the prevalence of risks associated with the full dimension of self-harm, suicidal ideation or attempts in LGBTQ+ young people who have these experiences.

2. To investigate whether there is a difference in the prevalence of risks between young people who identify as a sexual orientation minority (LGBQ) alongside those who identify as a gender identity minority (TGNC).

## Methods

### Protocol and registration

This review was conducted and reported in accordance with PRISMA guidelines (SM1) [29]. An a-priori protocol was registered on PROSPERO (CRD42019130037), and the full protocol was published in 2019 [30]. As this is a systematic review and meta-analysis of published literature, ethical approval was not sought.

### Search strategy

During March 2019, a literature search strategy was developed with an academic skills specialist at the University of Birmingham. An electronic search was conducted on the 31st of March 2019 using MEDLINE, Scopus, EMBASE, PsycINFO, and Web of Science. This was updated on the 31st of January 2020. There was no date limit for identified articles, however only those in English language were considered. Search terms (and their derivatives) focused on the variables of interest; "self-harm", "suic*", "adolescent*", "young person*", "sexual orientation", "gender identity" and "risk*", see Fig 1. The reference list of included articles and key papers within the field were examined for further relevant publications.

### Inclusion criteria

Articles included in this systematic review were empirical quantitative studies, which examined risks across the dimension of self-harm and suicide in LGBTQ+ young people (12–25

**Search strategy terms:**

(self-harm OR self harm* OR self-injur* OR "self injur*" OR self-cut* OR self-destruct* OR "self destruct*" OR "nonsuicidal self-injur*" OR "non-suicidal self injur*" OR "deliberate self harm" OR "deliberate self-harm" OR DSH OR "self-mutil*" OR overdos* OR self-inflicted injur* OR "self inflicted injur*" OR suicid* OR "parasuicid*" OR para-suicid* OR parasuicid* OR suicidal behav* OR suicide* OR "life-threatening behavio*" OR "suicide ideat*" OR "suicide attempt*" OR "attempted suicide*" OR NSSI)

AND

(moderat* OR mediat* OR "risk facto*" OR mechan* OR predict* OR pathway OR interact* OR "protective facto*" OR facto* OR influence OR correlate* OR precurs* OR "causal facto*")

AND

(transgender* OR transsexual* OR "gender nonconforming" OR "gender identity disorder" OR "gender dysphoria" OR "gender minority" OR lesbian*OR gay* OR bisexual* OR "sexual minority" OR "same-sex" OR homosexual* OR "homosexuality, male" OR "homosexuality, female" OR "gender identity" OR non-heterosexual* OR "non heterosexual*"OR homosexuality OR queer* OR questioning OR "non-binary" OR "non binary" OR "LGBT*" OR "sexual dissident*" OR "sexual and gender minorities" OR "gender variant" OR gender-variant OR genderqueer OR intersex OR "minority groups" OR "TGNC" OR "transgender and gender nonconforming")

AND

(Child* OR adolesc* OR "young people" OR kid* OR pupils OR youth OR juvenile OR "young adult*" OR "young person" OR minor*)

**Fig 1. Search strategy terms.**

years). This age range covers the period of adolescence and early adulthood [31]. An associated risk is operationalised as "an exposure that is statistically related in some way to an outcome" [32; p1], such as significant effect sizes, correlations, mediators, moderators, beta statistics, or any prevalence available relating to an outcome of self-harm or suicide. Mixed-method study designs were included if the quantitative aspects were relevant and extractable. Papers were included if they provided a self-reported or verified group who identified as a sexual orientation or gender identity minority, and any outcome of across the dimension of self-harm and suicide. Studies, whose population were not focused on any sexual orientation or gender identity minorities, were included if they presented information for LGBTQ+ participants separately or if authors were able to offer this information when contacted. Full inclusion criteria are described in Table 1.

## Study selection

The results of the systematic search are presented in Fig 2. Overall, the searches yielded 2457 results; 96 duplicates were removed. Studies were screened for eligibility at title, abstract and full-text by two independent researchers (AJW and AL) following the PRISMA guidelines [29]. Following the removal of duplications, 2361 were title and abstract screened. If agreement regarding the eligibility of an article could not be met through discussion, a third researcher (MM) was invited to review. This process was repeated at full-text screening for 465 articles,

**Table 1. Inclusion criteria used during screening process.**

| Inclusion Criteria | Exclusion criteria |
|---|---|
| • Peer reviewed studies.<br>• Any geographical location.<br>• English language.<br>• Empirical quantitative studies, following cross-sectional, prospective, longitudinal, cohort and case-control designs.<br>• Participants that have had a measured outcome from the dimension of self-harm and suicide; self-harm (self-harm or injury to self-irrespective of suicidal intent), suicidal ideation (thoughts, plan, death wish), or suicide attempt (individual took an attempt on their life, suicide death).<br>• Studies must consider risks associated with or predictive of self-harm, suicidal ideation, suicidal attempt or death.<br>• Participants must be young people (12–25 years).<br>• Participants that are identified or self-identified as any sexual or gender minority or member of LGBTQ+. | • Non-peer reviewed literature.<br>• Not English language.<br>• Grey literature such as theses, dissertations or conference proceedings.<br>• Articles such as commentaries, reviews, editorial or opinion pieces.<br>• Empirical qualitative studies.<br>• Participants who have no experience of self-harm, suicidal ideation or suicidal attempt.<br>• Sample not aged between 12 and 25 years, e.g. adults 26 years and above or children 12 years and under.<br>• Participants who are identified as heterosexual or not part of sexual or gender minority. |

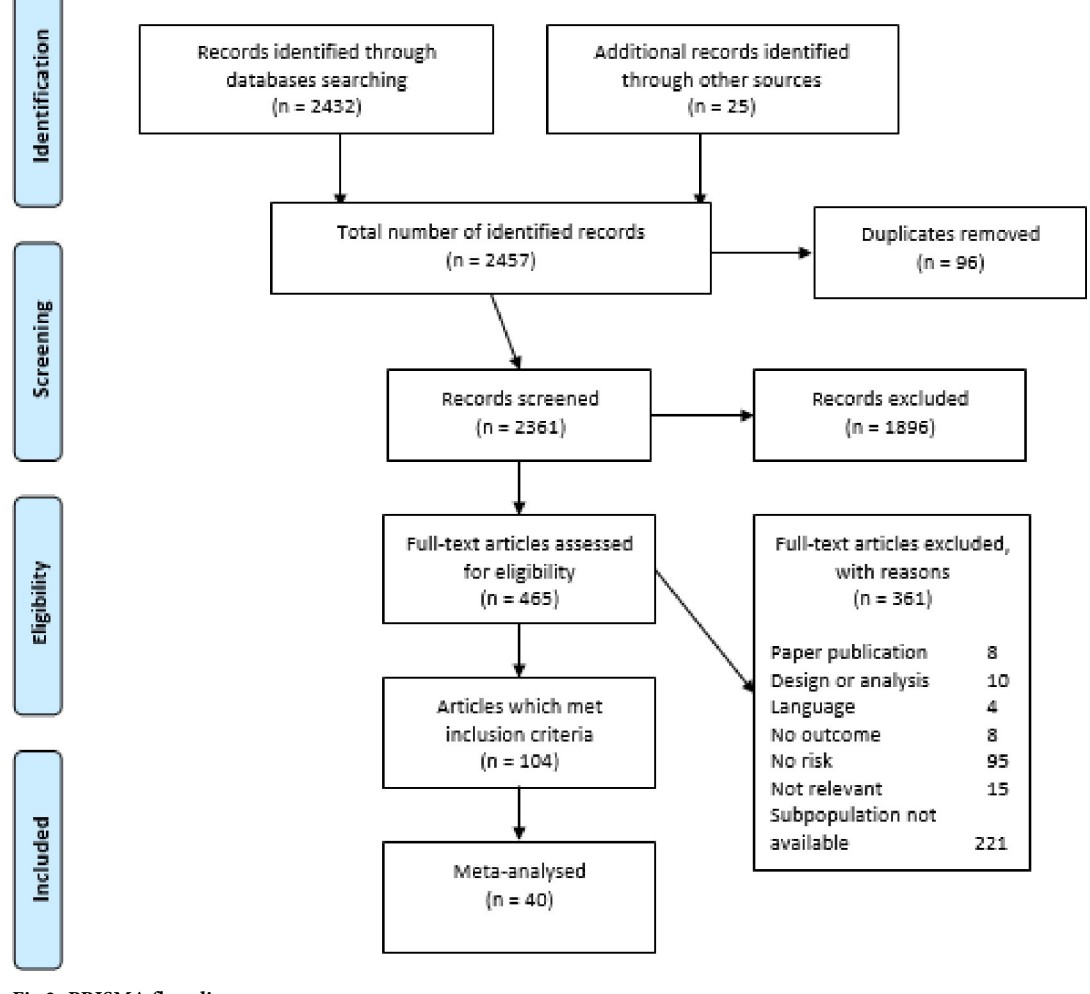

**Fig 2. PRISMA flow diagram.**

which produced a very high inter-rater reliability (Prevalence- And Bias-Adjusted Kappa, PABAK = 0.948) [33]. This was used due to PABAK being a more stable indicator of inter-rater reliability than Cohen's Kappa [34].

## Data extraction

A modified version of the data extraction tool used in a previous systematic review was utilised by two independent authors (AJW, AL) to extract data on study design, participants, outcome details, and associated risk [35]. After extraction was completed and checked, any disagreements were discussed and resolved by the research team. Risks were extracted based on a significant relationship to self-harm or suicide outcome. This has the potential to produce multiple reporting of the same study, as the risk may be reporting different outcomes for the same population or the same risk reported for multiple subgroups. For example, within one study, victimisation may be significantly associated with self-harm and suicidal ideation, both of which have an effect size. This would then be extracted twice to yield both sets of information. Initially, outcomes were combined into a single quantitative outcome [36]. Thereby, the overall prevalence of this risk for self-harm and suicide could be observed. Further analysis considered the risk to each outcome individually. The inclusion of multiple reporting from a single study may have resulted in a reduction in confidence intervals for the random effects model as the sample sizes will be included numerous times.

## Risk of bias assessment

To assess quality within the literature, variations of the Newcastle-Ottawa Scale (NOS) were employed [35–38]. This allowed a number of study designs to be considered and assessed. The forms assess risk of bias based on three core aspects of study design: participant selection, comparability of participants, and exposure ascertainment. These were adapted for this systematic review (see SM3), and rated as either low, moderate or high quality using the same category distinctions as previous research [39]. The two reviewers assessed the quality of studies independently, with intermediate agreement (PABAK = 0.43). Agreement was achieved through discussion.

## Data synthesis

The search strategy yielded 104 primary articles, across 102 studies. Given the large number of individual risk factors, similar variables were categorised resembling the format used by previous literature [40]; demographic, psychosocial, mental health difficulties. Rather than use "psychiatric or mental health" however, mental health difficulties was selected due to self-report measures commonly being used, the inclusion of symptomology, and limited information regarding diagnosis of mental health conditions. Additionally, two categories of risk were created, victimisation and LGBTQ+ specific risks. Victimisation includes individual measures which considered the process of the LGBTQ+ young person being treated poorly, harassed, abuse or discriminated against or subjected to bullying. LGBTQ+ specific risks included risks which were strongly related to the LGBTQ+ identity held by the young person, e.g. coming out stress [41], parent being unaware of sexual orientation [42], or negative attitudes towards homosexuality [43]. Risks were classed as victimisation if they suggested direct negative action against the individual, e.g. discrimination, bullying, harassment or threat. Victimisation was selected as representative title as it most often occurred within the studies. Risks which were both victimisation and LGBTQ+ specific, such as trans, bi, and homophobic bullying, were categorised as victimisation.

There was a large amount of inconsistency among individual risks for three categories: demographic, psychosocial and LGBTQ+ specific risks. This did not allow for meaningful clustering of variables into meta-analysis which would provide a prevalence of risk among LGBTQ + young people who had experiences of self-harm or suicide. Furthermore, numerical evidence was not available for many individual risks; in these instances, either there was no statistically significant statistics available for associated risks, effect sizes, correlations, mediators, moderators, beta statistics, or any reporting of prevalence. Numerical data was predominantly available within victimisation and mental health difficulties; therefore these risks were analysed. The 65 studies not included in meta-analysis due to are briefly described by risk category, and separated by population (e.g. sexual orientation minority, gender identity minority, LGBTQ + umbrella).

## Numerical analysis

A meta-analysis was conducted for two risks associated with self-harm and suicide among LGBTQ+ young people; victimisation and mental health difficulties, where sufficient data for aggregation were available. For these two risks, outcome data from forty primary studies were synthesised. The purpose of the meta-analysis was to 1) to investigate the prevalence of victimisation and mental health difficulties associated with self-harm, suicidal ideation or suicidal attempt among LGBTQ+ young people with these experiences; 2) to investigate whether there is a difference in the prevalence of victimisation and mental health difficulties among those young people who identify as a sexual orientation minority (LGBQ) and those who identify as a gender identity minority (TGNC); 3) to identify whether the prevalence of victimisation and mental health difficulties is different in LGBTQ+ young people who have experiences of self-harm, suicidal ideation or attempt compared with cisgender heterosexual young people with these experiences.

Event rates of primary studies were log transformed before numerical syntheses such that they were all the same unit of measure (but back-transformed for clear presentation in tables). Studies with an event rate of zero or one were excluded from analysis as studies with a small sample size do not permit accurate estimations of event rate. Where data was available for the target population subgroup and a control subgroup of cisgender and heterosexual individuals, odds ratios were calculated.

The random effects model was used as this assumes that not all studies have the same power to detect effects, therefore, a common effect size cannot be assumed. As the study effects were normally distributed, the DerSimonian and Laird method was selected to determine the variation between the studies to fit the random effects model [44]. The random effects model was extended to include explicit consideration of the methodological quality of the primary studies. This "quality effects model" (QEM) used the NOS total score to characterise the overall quality of the study. This QEM model can be interpreted as the meta-analytic synthesis that would have been obtained if all the studies had been of the same methodological quality as the highest rated study within the review, thereby providing a measure of attenuation to the methodological variation of included studies.

Higgins $I^2$ was used to determine the level of heterogeneity within the primary studies with a value of above 75% considered problematic. Sensitivity analysis was conducted to identify studies disproportionately influencing results. Such studies were excluded from subsequent analyses due to the high risk of bias. Subgroup analysis was also used to aid the identification of sources of problematic heterogeneity.

Publication bias and small study effects were also estimated by inspection of funnel plots. In absence of publication bias, high precision studies will be evidenced near the average, with lower precision studies spread evenly and symmetrically on both sides of the average, creating

a funnel-shaped distribution. Publication bias is indicated by the absence of studies in the area of the final plot associated with small (i.e. non-significant) effect sizes in small studies.

If publication bias was evidenced then a trim and fill procedure was undertaken. This produced an adjusted effect size (controlling for publication bias), and the impact of publication bias was assessed by comparison with the uncorrected random effects model. The fail-safe N was also calculated using the Orwin algorithm [45]. This is the estimation of missing studies that was required to render the effect non-significant. If the fail-safe N is large (in relation to the number of studies included in the synthesis), then the synthesis could be considered robust to the effects of publication bias.

Before searches were conducted, two a-priori hypotheses were established to consider heterogeneity which may occur within the data [30]. The first suggested that heterogeneity may be explained by consideration of sexual orientation (LGBQ) and gender identity minorities (TGNC) as separate populations. This allows us to determine whether there are similar levels of risk within both groups. The second a-priori aim was to consider risk by age group; however, this was not possible given the final dataset. Additionally, a subgroup analysis was run based on the type of outcomes reported: self-harm, suicidal ideation, and suicidal attempt. Summary effects and associated heterogeneity measures were calculated for each subgroup, the significance of difference between these being evaluated by the comparison of their 95% confidence intervals.

## Results

One-hundred and four papers from 102 studies were included, which met all the inclusion criteria and contained extractable significant risks associated with self-harm, suicidal ideation, or suicidal attempt. Twenty-six studies examined a form of self-harm (e.g. self-harm with suicidal intent, self-harm intent unspecified, non-suicidal-self-injury) whereas 77 considered ideation and 76 considered behaviour, studies often considered more than one outcome. None of the studies included information on participants who died by suicide. Two of the included papers [46,47] utilised the same dataset as a previously included study [48,49]. These were included as separate papers, given that they highlight risk factors which the primary study did not. The majority of studies were cross-sectional (n = 91); with 10 longitudinal studies, and 3 cohort studies. A total of 1,146,395 participants were included, with 129,469 (11.3%) being LGBQ and 13,041 (1.1%) being TGNC. Ages ranged from 12–25 (M = 17.7, SD = 1.9). Studies were mainly based within the U.S.A (n = 77), followed by the U.K. (n = 7), and China (n = 4). For full individual study characteristics, see supplementary materials 4 tables A [17,18,41,43,49–109] and B [16,23,42,47,110–144] (SM3) in S1 Table. From this document, further figures regarding heterogeneity and influential studies are also available.

From the 104 included papers, 64 were unable to be numerically synthesised [17,18,41,43,49–109]. The individual characteristics of these studies can be seen in Supplementary Table A (SM3) in S1 Table. The population of these papers represented a total of 929,802 individuals, of whom 90,767 were LGBTQ+ identifying (9.76%). Therefore, these studies are considered 81.1% of the overall population. These studies did evidence multiple risks associated with experiences of self-harm and suicide among LGBTQ+ young people. The individual risk factors were varied and numerous to the extent that they could not be individually considered in relation to prevalence. However, by categorising these broadly, some information can be gained.

Most of the papers which were not numerically synthesised, focused on samples which only considered sexual orientation minorities, see Table 2. With fewer studies examining TGNC populations or across the LGBTQ+ umbrella. Across all populations, psychosocial risks were

**Table 2. Risks associated with experiences of self-harm or suicide among LGBTQ+ young people: Data unable to be numerically synthesised.**

| Categories of risk | LGBQ k = 48 N (%) | TGNC k = 8 N (%) | LBGTQ+ k = 8 N (%) |
|---|---|---|---|
| **Demographic variables** (*e.g. natal gender, age, race*) | 15 (30.6) | 4 (50) | 3 (37.5) |
| **Psychosocial variables** (*e.g. low self-esteem, dating violence, suicide of friend or family, abuse*) | 31 (63.3) | 4 (50) | 5 (62.5) |
| **Victimisation variables** (*e.g. LGBTQ hate crime, homophobic bullying, school bullying, cyber bullying*) | 27 (55.1) | 2 (25) | 4 (50) |
| **Mental health difficulties variables** (*e.g. depression, substance use, bipolar, anxiety*) | 10 (20.4) | 4 (50) | 2 (25) |
| **LGBTQ+ specific variables** (*e.g. gender-role nonconformity, internalised homophobia, parental rejection, loss of friends due to sexual orientation*) | 13 (26.5) | 2 (25) | 3 (37.5) |

most commonly cited in associated with self-harm and suicide. Victimisation and mental health difficulties were evident, although without reinforcing numerical evidence.

## 1. Meta-analysis: Victimisation

A random effects model was calculated, using the generic inverse variance method, to examine the prevalence of victimisation as a risk associated with experiences of self-harm, suicidal ideation or suicidal attempt among LGBTQ+ young people. Sixty-three estimates from 31 individual samples were reported, representing 331,321 participants in total. The random effects models reported a pooled prevalence estimate of 0.33 and a 95% confidence interval of between 0.29–0.38 among LGBTQ+ young people with self-harm or suicide experiences.

A high level of between study variation (heterogeneity) could not be attributed to differences in individual reaction to victimisation within the included studies (Higgin's $I^2$ = 99%). Therefore, the prevalence estimates of the primary studies may be influenced by the presence of uncontrolled or confounding factors. Given this substantial level of heterogeneity, the impact of disproportionately influential individual studies was assessed using a leave-one-out analysis. Following this, Taliaferro and Muehlenkamp (2017) [137] was removed from the meta-analysis [137]. This was due to a variable being extracted multiple times as numerical data was given per sexual orientation, this resulted in a large volume of included variables. Therefore, this study was overtly overrepresented within the sample.

The random effects model was recalculated with 55 measures of prevalence from 30 unique samples. The corrected random effects model reported a pooled prevalence estimate of 0.36 (95%CI: 0.31–0.40) (Fig 3). The corrected random effects model did not impact heterogeneity (Higgin's $I^2$ = 99%). Accordingly, the observed heterogeneity could not be considered to be the result of overly influential individual studies, and therefore other sources of heterogeneity require exploration.

The Quality Effects Model was calculated using the total score from the risk of bias ratings, (individual study ratings can be found in SM3). The QEM can be interpreted as the meta-analytic synthesis that would have been obtained had all the studies been of the same methodological quality as the best study within the review. This reported an estimate of 0.36 (95% CI: 0.31–0.41). Given the similarity between the random effects model and the synthesis derived from the quality effects model, it is possible to conclude that the ratings of methodological quality did not have a significant and substantial impact upon the estimates of prevalence.

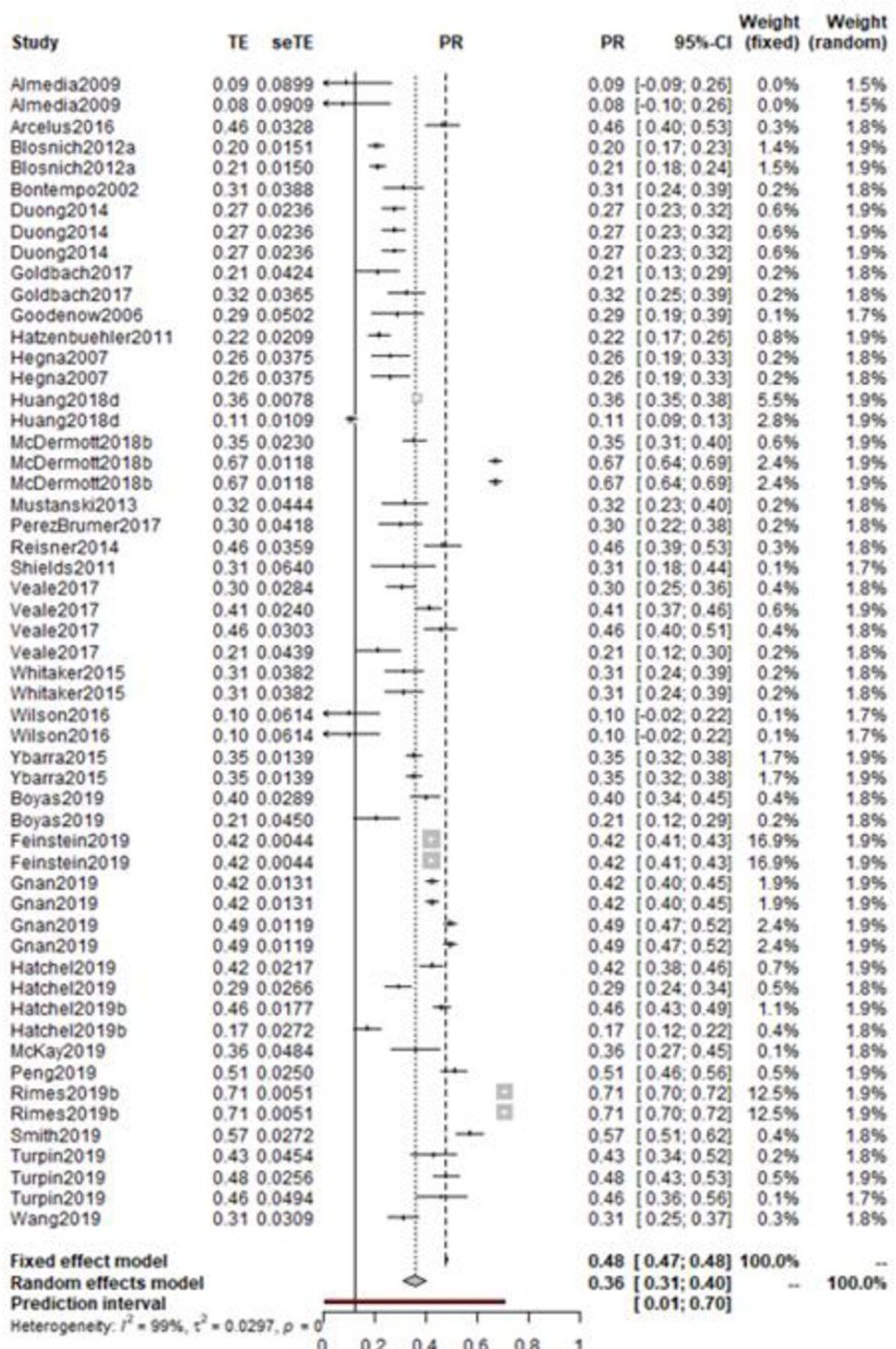

**Fig 3. Forest plot of victimisation prevalence among LGBTQ+ with experiences of self-harm or suicide.**

Visual inspection of the funnel plot of victimisation prevalence there is little evidence of publication bias. A fail-safe number of 107 suggested that an additional 101.9% of the existent literature would be required for unpublished null effects for the meta-analytic effect to become non-significant. Thus, the observed effect is considered robust to publication bias.

**Table 3. Subgroup analyses of victimisation prevalence among LGBTQ+ young people with self-harm or suicidal experiences.**

| | Number of estimates (N) | Prevalence Rate | 95% CI | Q | $I^2$ (%) | $\Sigma^2$ | Q, df, p |
|---|---|---|---|---|---|---|---|
| **QUALITY RATING** | | | | | | | Q = 19.50, df = 2, p = 0.01 |
| **Low** | 7 | 0.46 | 0.34–0.58 | 347.88 | 98.3 | 0.02 | |
| **Moderate** | 31 | 0.28 | 0.24–0.32 | 686.32 | 95.6 | 0.01 | |
| **High** | 17 | 0.45 | 0.37–0.52 | 4107.33 | 99.6 | 0.02 | |
| **POPULATION** | | | | | | | Q = 0.11, df = 1, p = 0.74 |
| **LGBQ** | 27 | 0.34 | 0.27–0.42 | 6282.68 | 99.6 | 0.03 | |
| **TGNC** | 9 | 0.33 | 0.24–0.41 | 108.99 | 92.7 | 0.01 | |
| **OUTCOME** | | | | | | | Q = 12.18, df = 2, p = 0.01 |
| **Self-harm** | 10 | 0.39 | 0.31–0.48 | 353.09 | 97.5 | 0.02 | |
| **Suicidal ideation** | 21 | 0.35 | 0.33–0.38 | 212.38 | 93.4 | 0.00 | |
| Suicidal attempt | 15 | 0.26 | 0.20–0.31 | 212.38 | 93.4 | 0.01 | |

To further assess the impact of methodological variation upon heterogeneity, a series of subgroup analyses were conducted (Table 3). The first considered risk of bias ratings; low, moderate, and high quality (Q = 19.5, p < 0.01). Both high-rated and low-rated studies evidenced higher prevalence than those rated as moderate quality.

Subgroup analysis was utilised to explore the impact of uncontrolled covariates upon victimisation. Initially, this evaluated differences in prevalence of victimisation between groups of sexual orientation (LGBQ) or gender identity groups (TGNC) with these experiences of self-harm and suicide. This analysis was to explore whether a particular identity group experiences greater victimisation than others. Studies which combined the populations or looked at just one representation of LGBQ were excluded from this analysis. The subgroup analysis showed that prevalence rates of victimisation were relatively consistent across all gender identity and sexual orientation studies/groups (Q = 0.11, p = 0.74). However, heterogeneity was notably lower within the TGNC studies. This may be related to a small number of studies being included, as analysis of LGBQ triples the study sample. Following this, subgroup analysis was conducted regarding outcome. Again, studies were excluded if they collapsed two distinct categories; suicidal ideation and suicidal attempt. Studies with self-harm as outcome demonstrated an overall victimisation prevalence rate of 39%. This suggests that higher rates of victimisation are associated with self-harm when compared to suicidal thoughts or attempts among LGBTQ+ participants.

The prevalence of victimisation within LGBTQ+ young people with experiences of self-harm or suicide was compared to matched cisgender, heterosexual control counterparts. These young people also had experiences of self-harm or suicide. The odds ratios (19 estimates from 12 studies) were synthesised using the generic inverse variance. An odds ratio of 4.82 (CI: 3.67–6.32) was reported. Between studies heterogeneity was high ($I^2$ = 98%) suggesting uncontrolled methodological or conceptual factors contributing variations in reported risks. Therefore, a leave-one-out analysis was conducted to identify studies that might be exerting a disproportionate influence on the overall meta-analysis. One study was identified as both heterogeneous and influential, demonstrated by a change of effect of over 13%. Thus, Turpin and colleagues' study was removed to give a more conservative overall odds ratio [139].

The following meta-analysis was based on the remaining 16 odds ratios from 12 studies. This produced a synthesised odds ratio of 3.74 (95% CI: 2.90–4.84)(Fig 4). The corrected random effects model produced very little change to the heterogeneity level, (Higgin's $I^2$ = 98%). Given the small number of studies, further analyses including an assessment for publication bias were not feasible.

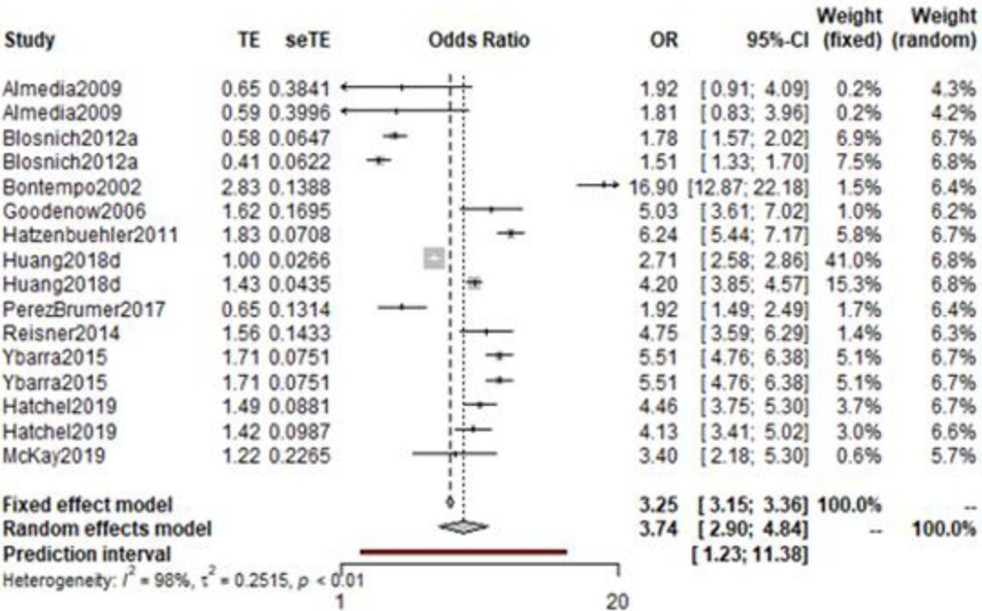

**Fig 4. Odds ratio among LGBTQ+ young people with experiences of self-harm or suicide compared to cisgender, heterosexual young people with experiences of self-harm or suicide.**

## 2. Meta-analysis: Mental health difficulties

A second random effects model was calculated to consider the prevalence of previous mental health difficulties within LGBTQ+ young people who have an experience of self-harm, suicidal ideation or suicidal attempt. A total of 166,810 participants were assessed over 22 studies which produced 51 estimates. The model calculated a prevalence of mental health difficulties of 0.36 (95% CI: 0.29–0.43). Again, a high level of heterogeneity was found (Higgin's $I^2$ = 99%). A leave-one-out analysis was therefore run, with the influential studies being evaluated for inclusion. Studies were omitted if they disproportionally influenced the overall result [136–138]. The random effects models were then recalculated with the 19 studies and 32 variables. This resulted in the prevalence of mental health difficulties increasing to 0.39 (95% CI: 0.31–0.47) (Fig 5). While high heterogeneity remained (Higgin's $I^2$ = 98%).

Visual observation of a funnel plot and trim-and-fill procedure suggests the absence of publication bias. Following Orwin's algorithm, it was shown that 31 unpublished null studies would be needed to reduce the meta-analytic effect found within this sample. Again, subgroup analyses considering the risk of bias were conducted. The QEM model reported an estimate of 0.39 (95% CI: 0.31–0.47), suggesting that there were not enough differences regarding the risk of bias ratings to substantially influence the overall effects. Subgroup analysis of this sample demonstrated that 4 studies were considered high quality, 14 were of moderate quality and 3 of low quality. However, little could be concluded from between groups differences (Q = 1.54, P = 0.46).

Further subgroup analyses were conducted to investigate the impact of uncontrolled covariates relating to mental health difficulties prevalence (Table 4). The first of these again considered the prevalence differences which may occur between LGBQ and TNGC samples. This analysis evidenced that LGBQ young people were shown to have a higher prevalence of mental health difficulties than TGNC individuals (42% vs 34%). The difference in effect size is likely related to the large difference of included studies. The Higgins $I^2$ value for both groups were still high, suggesting that these studies do contribute to heterogeneity, although to lesser extent

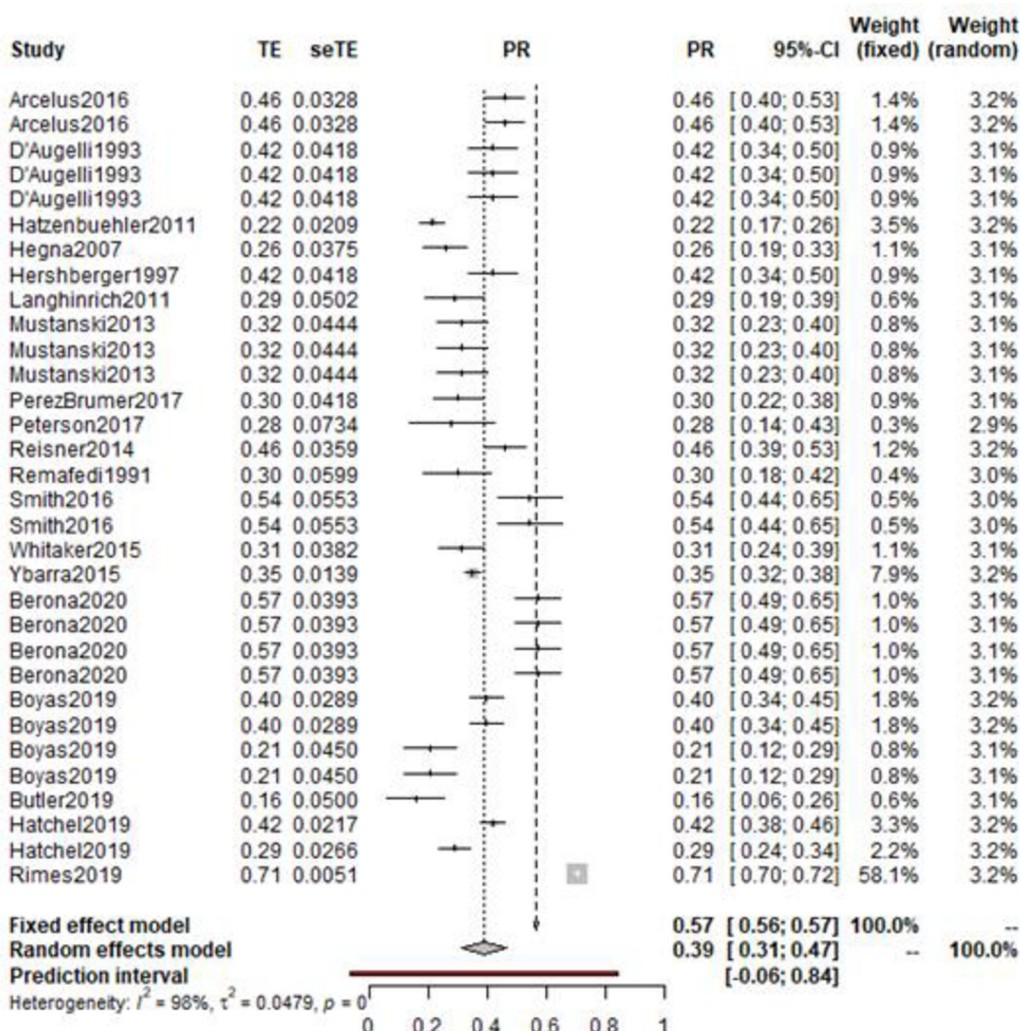

**Fig 5. Overall prevalence of mental health difficulties within LGBTQ+ young people with experiences of self-harm or suicide.**

**Table 4. Subgroup analyses of mental health difficulties prevalence among LGBTQ+ populations who have experiences of self-harm or suicide.**

| | Number of estimates (N) | Prevalence Rate | 95% CI | Q | I² (%) | Σ² | Q, df, p |
|---|---|---|---|---|---|---|---|
| **QUALITY RATING** | | | | | | | Q = 1.54, df = 2, p = 0.46 |
| **Low** | 11 | 0.41 | 0.33–0.49 | 122.06 | 91.8 | 0.01 | |
| **Moderate** | 17 | 0.36 | 0.31–0.41 | 125.83 | 87.3 | 0.00 | |
| **High** | 4 | 0.47 | 0.25–0.69 | 417.38 | 99.3 | 0.05 | |
| **POPULATION** | | | | | | | Q = 2.43, df = 1, p = 0.30 |
| **LGBQ** | 20 | 0.42 | 0.32–0.53 | 1227.71 | 98.5 | 0.05 | |
| **TGNC** | 5 | 0.34 | 0.22–0.45 | 37.56 | 89.4 | 0.01 | |
| **OUTCOME** | | | | | | | Q = 0.41, df = 2, p = 0.82 |
| **Self-harm** | 3 | 0.38 | 0.20–0.53 | 30.19 | 93.4 | 0.02 | |
| **Suicidal ideation** | 8 | 0.40 | 0.35–0.44 | 32.70 | 78.6 | 0.00 | |
| Suicidal attempt | 19 | 0.38 | 0.31–0.44 | 222.21 | 91.9 | 0.02 | |

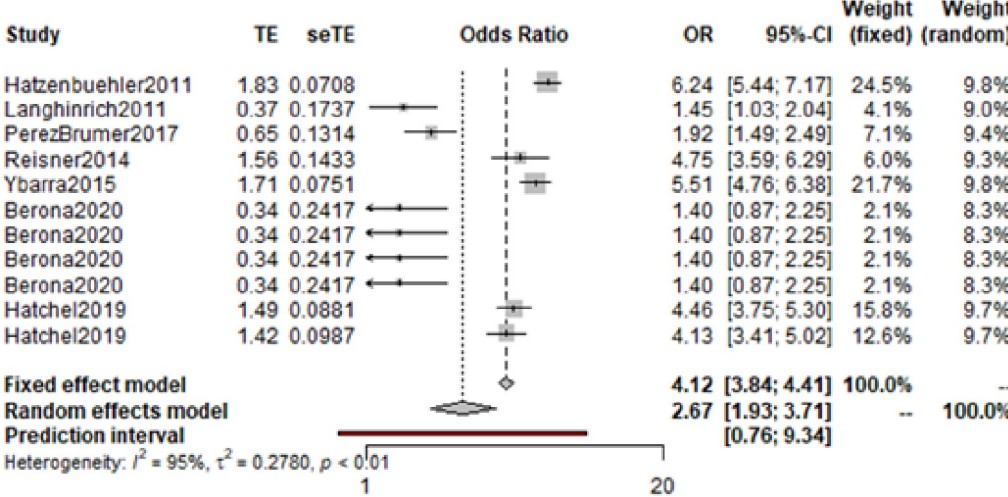

**Fig 6. Odds ratio of LGBTQ+ young people with experiences of self-harm or suicide compared to cisgender, heterosexual peers with experiences of self-harm or suicide.**

within TGNC populations. A similar subgroup analysis regarding outcome was conducted, this demonstrated that the rates of mental health difficulties were slightly more prevalent among those with suicidal ideation.

Following this, a meta-analysis of odds ratios was conducted; considering prevalence of mental health difficulties among LGBTQ+ young people and cisgender, heterosexual young people both with experiences of self-harm or suicide. Only 7 studies had available data. The random effects model calculated an odds ratio of 2.67 (95% CI: 1.93–3.71), with a high level of heterogeneity ($I^2$ = 95%) (Fig 6). However, due to the limited number of studies, further analysis was not conducted.

## Discussion

This is the first meta-analysis which evidences prevalence of victimisation and mental health difficulties within young people aged 12–25 who identify as LGBTQ+ with experiences of self-harm, suicidal ideation and attempt. The review consisted of 142,510 participants who were a sexual orientation or gender identity minority. Due to limited information reported within the studies, it was not possible to consistently consider TGNC participants by their sexual orientation as well. Evidence demonstrated high prevalence of victimisation (36%) and mental health difficulties (39%) within these populations. Our review shows that these experiences were respectively 3.74 times and 2.67 times higher among young LGBTQ+ people than their cisgender, heterosexual counterparts. There were only 10 studies which were considered high-quality, with most studies (81%) being rated as moderate quality. Substantial heterogeneity was observed between study estimates within both meta-analyses.

The key findings of this meta-analysis strongly support previous research [9,20,22–26]. Within this study, a broad view of victimisation was arrogated, including a range of bullying behaviours such as cyber victimisation, homophobic bullying, peer bullying and so forth. Preceding meta-analyses have previously demonstrated established links between peer victimisation and suicide and LGBT victimisation and non-suicidal self-injury (NSSI) [25,26]. This review demonstrates that there is a high prevalence between LGBTQ+ young people experiencing various forms of victimisation and self-harm and suicide. Indeed, this link

between victimisation and self-harm and/or suicide appears to be more common than that among cisgender, heterosexual peers.

Mental health difficulties were also shown to be highly prevalent with self-harm and suicide among LGBTQ+ young people. Liu and colleagues also evidenced mental health difficulties were linked to NSSI within this population [25]. The current review extends findings from previous research by calculating risk prevalence and odds across the spectrum of self-harm to suicide and differentiating by gender identity and sexuality [25,26]. Thus, demonstrating that higher rates of victimisation and mental health difficulties are found in LGBTQ+ young people who experience self-harm and suicide. However, evidence is not available from this review as the causal pathway causing self-harm or suicide or how predictive these risks associated with self-harm and suicide are.

By looking across the broad umbrella LGBTQ+ identities, this review has assessed the prevalence of risks associated with self-harm and suicide by gender identity compared to sexual orientation minorities groups. This allows for consideration of how influential these risks might be to particular groups among the LGBTQ+ label, and where differences of risk may lie. Both victimisation and mental health difficulties were evidenced to be more prevalent within LGBQ young people rather than TGNC. However, it is likely that our finding is due to the higher number of studies focusing solely on LGBQ populations, as noted by the wider confidence intervals seen within the TGNC subgroup analyses. Furthermore, those studies which considered both sexual orientation and gender identity, tend to have low numbers of TGNC participants. Therefore, the TGNC risks are potentially conflated or ignored, as there is a lack of statistical power to evidence risks which may apply to TGNC participants and not LGBQ.

Further to this, we were unable to conduct meta-analysis by identity (e.g. transgender man, transgender woman, nonbinary etc.) within gender identity or sexuality (e.g. bisexual, homosexual, lesbian), thereby these are broadly categorised. This may overlook differences between identifying as a particular sexual orientation or gender identity; and, how being a member of these subgroups may differ from each other [145]. Additionally, no papers considered sexualities outside of homosexual, bisexual, queer or questioning. This limits how far these risk conclusions might be drawn to other sexual orientation groups e.g. those who are asexual, pansexual, polysexual etc. Future research should support inclusion of diverse sexualities and gender identities within studies, offering individuals to self-report in their own words, and options for intersectional identities.

This review has important clinical and policy implications in relation to suicide prevention among LGBTQ young people. Primarily, discrimination against LGBTQ+ individuals has widely been recognised as a priority for governments and organisations globally [146,147]. These results definitively highlight the harmful outcomes associated with acts of discrimination and victimisation. Given the variety of countries which are included in this study, the findings of this study could be used to inform national policies, such that there is a priority focus on reducing minority victimisation and discrimination. Furthermore, by understanding these complex experiences which surround LGBTQ+ youth, compounded by high rates of mental health difficulties, suicide prevention strategies are better informed to support LGBTQ + youth. Thereby suicide prevention interventions and policies may be better tailored to the specific needs of LGBTQ+ young people and develop initiatives which build resilience and challenge societal acceptance of such discrimination. However, the studies in this meta-analysis mainly come from High-Income Countries (HIC), therefore the results might not be generalisable to Low- and Middle- Income Countries (LMIC) where young people who identify as LGBTQ+ may face additional or different types of risks.

Secondly, health care professionals should be aware of the high prevalence of mental health difficulties and victimisation within the umbrella of LGBTQ+ young people. Acknowledging

sexuality and gender identity in an accepting and supportive manner, would be beneficial to encouraging a constructive health care environment [148,149], which could potentially aid disclosure of self-harm and suicide. Evidence also shows that health professionals encouraging LGBTQ+ youth to discuss their experiences of victimisation could further reduce negative health consequences [150]. From these insights, professionals might be able to suggest treatments or care understanding the sociodemographic environment which these individuals are living in.

Much of this research takes places within school settings, with the average age of participants being below 18 years old. Given that bullying among school-aged children is common [151], this would suggest that school-based interventions would be an appropriate setting to target victimisation for LGBTQ+ young people, potentially reducing self-harm and suicide. This is supported by a recent study suggesting that addressing the barriers and facilitators when reporting and responding to LGBTQ+ victimisation in schools would prevent adverse mental health [152]. In particular, LGBTQ+ youth felt that building trust with staff members, being given time to discuss problems and receiving responses from school were key [152]. Therefore, creating an environment which recognises the unique aspects and potential risks of being LGBTQ+, such as dealing with difficult disclosure [118] or understanding gender nonconformity [25] would be beneficial. This could translate to older adolescents and young adults by having similar environments within colleges, universities or social community spaces. These spaces might be able to consider risks, which differentiate by age (e.g. identity development, transition treatments available, housing situations) which due to limited reporting we were unable to meta-analysis within this review.

There is a wealth of literature readily available relating to risks for self-harm and suicide within LGBTQ+ young people. However importantly, even though many of these studies had explicit focus on LGBTQ+ individuals, only 12% of the total population held these identities and reporting is highly inconsistence between individual risks. Future research in the field of self-harm and suicide prevention requires a specific LGBTQ+ focus as this would allow for a holistic understanding of these populations' experiences.

## Strengths & limitations

This is the first systematic review and meta-analysis which has comprehensively synthesised existing evidence from across the full spectrum of LGBTQ+ young people in order to identify the prevalence of key risks with self-harm and suicide. Firstly, this dimensional approach allowed for a holistic view and comparison of risk prevalence across self-harm and suicidal thoughts and behaviours. Secondly, broad search strategies were run, which ensured a large amount of studies was identified across disciplines and across the LGBTQ+ umbrella. This search was re-run prior to submission to ensure that the review was as up-to-date as possible. Thereby, TGNC populations were able to be identified and specifically examined with reference to similar LGBQ samples. A final strength was the robust meta-analytic strategy which was emplaced within this study, therefore allowing authors to determine points of bias and control for these.

There were, however, some limitations which need to be considered. Firstly, there were few high-quality studies and substantial heterogeneity within the findings. Sources of heterogeneity were explored using our pre-specified subgroup analysis but also to determine points of heterogeneity; this offered little. Potentially, this was related to the use of four variations of the NOS assessment (see SM2). While inclusion of four versions allowed for a greater number of papers to be assessed, this also created another variable of ambiguity. However, heterogeneity may also be related to the variation in conceptualisation of phenomena, population, study design and fundamentally individual reporting of risk. In future, clear operationalisation

within studies is necessary and use of standardised, validated measures to assess self-harm and suicide across the spectrum of thoughts and behaviours.

Secondly, self-harm with suicide intention and self-harm without suicide intention may have different associated risks which link to why someone might be more likely to consider suicide. However, given the measures used to assess self-harm within included studies this was not possible. Therefore, only risks associated with self-harm regardless of intention was able to be analysed. This does not allow us to offer explanation as to why someone might consider suicide with this behaviour. Finally, searches were limited to English language; thereby key studies within other languages may have been overlooked.

## Supporting information

**S1 Fig. Search strategy terms.**
(TIF)

**S2 Fig. PRISMA flow diagram.**
(TIF)

**S3 Fig. Forest plot of victimisation prevalence among LGBTQ+ with experiences of self-harm or suicide.**
(TIF)

**S4 Fig. Odds ratio among LGBTQ+ young people with experiences of self-harm or suicide compared to cisgender, heterosexual young people with experiences of self-harm or suicide.**
(TIF)

**S5 Fig. Overall prevalence of mental health difficulties within LGBTQ+ young people with experiences of self-harm or suicide.**
(TIF)

**S6 Fig. Odds ratio of LGBTQ+ young people with experiences of self-harm or suicide compared to cisgender, heterosexual peers with experiences of self-harm or suicide.**
(TIF)

**S1 Table. Inclusion criteria used during screening process.**
(DOCX)

**S2 Table. Risks associated with experiences of self-harm or suicide among LGBTQ+ young people: Data unable to be numerically synthesised.**
(DOCX)

**S3 Table. Subgroup analyses of victimisation prevalence among LGBTQ+ young people with self-harm or suicidal experiences.**
(DOCX)

**S4 Table. Subgroup analyses of mental health difficulties prevalence among LGBTQ+ populations who have experiences of self-harm or suicide.**
(DOCX)

**S1 File. PRISMA checklist.**
(DOC)

**S2 File. NOS.**
(PDF)

**S3 File. Supplementary results.**
(DOCX)

## Author Contributions

**Conceptualization:** A. Jess Williams, Jon Arcelus, Ellen Townsend, Maria Michail.

**Data curation:** A. Jess Williams.

**Formal analysis:** A. Jess Williams, Christopher Jones.

**Investigation:** A. Jess Williams.

**Methodology:** A. Jess Williams, Aikaterini Lazaridou, Maria Michail.

**Project administration:** A. Jess Williams.

**Software:** Christopher Jones.

**Supervision:** Jon Arcelus, Ellen Townsend, Maria Michail.

**Validation:** A. Jess Williams.

**Visualization:** A. Jess Williams.

**Writing – original draft:** A. Jess Williams.

**Writing – review & editing:** A. Jess Williams, Christopher Jones, Jon Arcelus, Ellen Townsend, Maria Michail.

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
