## [Decision Letter · Decision Letter 0]

30 Nov 2020

PONE-D-20-34056

A systematic review and meta-analysis of victimisation and mental health prevalence among LGBTQ+ young people with experiences of self-harm and suicide.

PLOS ONE

Dear Dr. Williams,

Thank you for submitting your manuscript to PLOS ONE. After careful consideration, we feel that it has merit but does not fully meet PLOS ONE’s publication criteria as it currently stands. Therefore, we invite you to submit a revised version of the manuscript that addresses the points raised during the review process.

We look forward to receiving your revised manuscript.

Kind regards,

Vincenzo De Luca

Academic Editor

PLOS ONE

Journal Requirements:

2. Please include the full electronic search strategy in the Methods section of the manuscript.

3. Thank you for stating the following at the end of your manuscript:

'FUNDING STATEMENT

This work was supported by the Economic and Social Research Council for funding this research. The funder of the study had no role in study design, data collection, data analysis, data interpretation, or writing of the report. The corresponding author had full access to all the data in the study and had final responsibility for the decision to submit for publication.'

'No - The funders had no role in study design, data collection and analysis, decision to publish, or preparation of the manuscript.'

Reviewers' comments:

Reviewer's Responses to Questions

**Comments to the Author**

1. Is the manuscript technically sound, and do the data support the conclusions?

Reviewer #1: Yes

2. Has the statistical analysis been performed appropriately and rigorously? 

Reviewer #1: Yes

3. Have the authors made all data underlying the findings in their manuscript fully available?

Reviewer #1: Yes

4. Is the manuscript presented in an intelligible fashion and written in standard English?

Reviewer #1: Yes

5. Review Comments to the Author

Reviewer #1: This is an interesting meta-analysis on socially and clinically relevant issues on the border of psychiatry and psychology. In my opinion, the presented analysis was carried out correctly and the methodology has been described sufficiently. The authors performed a careful review according to the PRISMA guidelines and the registered and previously published study protocol. The study was carefully thought out, thanks to which the set goals were achieved. The description of the results is correct, and the discussion interesting and, like the other parts of this article, they do not require major corrections.

I would only suggest to include some information (in the introduction) on the global suicide rate among young people (generally, regardless of orientation) and their causes (e.g. according to WHO reports). In addition, please add keywords that were used in the database search.

6. PLOS authors have the option to publish the peer review history of their article (what does this mean?). If published, this will include your full peer review and any attached files.

Reviewer #1: No

---

## [Author Response · Author response to Decision Letter 0]

8 Dec 2020

Response to reviewers:

Thank you for highlighting these changes, this has now been completed.

2. Please include the full electronic search strategy in the Methods section of the manuscript.

This has now been added see figure 1, search strategy.

3. Thank you for stating the following at the end of your manuscript:

'FUNDING STATEMENT

This work was supported by the Economic and Social Research Council for funding this research. The funder of the study had no role in study design, data collection, data analysis, data interpretation, or writing of the report. The corresponding author had full access to all the data in the study and had final responsibility for the decision to submit for publication.'

'No - The funders had no role in study design, data collection and analysis, decision to publish, or preparation of the manuscript.'

This statement has now been removed from the manuscript.

This statement has now been removed from the manuscript. We would be happy to have the following funding statement;

“This project was funded as part of an Economic and Social Research Council grant on the Doctoral Training Pathway. The lead author, A. Jess Williams, receives a student stipend from the ESRC. The funders had no role in study design, data collection and analysis, decision to publish, or preparation of the manuscript.”

I appreciate this, I have included this in the cover letter.

This has now been completed, thank you for highlighting this.

Reviewer comment

Reviewer #1: This is an interesting meta-analysis on socially and clinically relevant issues on the border of psychiatry and psychology. In my opinion, the presented analysis was carried out correctly and the methodology has been described sufficiently. The authors performed a careful review according to the PRISMA guidelines and the registered and previously published study protocol. The study was carefully thought out, thanks to which the set goals were achieved. The description of the results is correct, and the discussion interesting and, like the other parts of this article, they do not require major corrections.

I would only suggest to include some information (in the introduction) on the global suicide rate among young people (generally, regardless of orientation) and their causes (e.g. according to WHO reports). In addition, please add keywords that were used in the database search.

Thank you for your kind review. We really appreciate the time and effort which has been taken to review this manuscript. 

• Per your suggestion we have added information regarding adolescent suicide rates; lines 51-52.

• Risks associated with young people generally was previously included, however we have made this clearer following your comments; lines 64-69.

• Search terms and derivatives have been included within the search strategy section; lines 112-114 and we have highlighted that for full search term details readers can view figure 1.

---

## [Decision Letter · Decision Letter 1]

28 Dec 2020

A systematic review and meta-analysis of victimisation and mental health prevalence among LGBTQ+ young people with experiences of self-harm and suicide.

PONE-D-20-34056R1

Dear Dr. Williams,

We’re pleased to inform you that your manuscript has been judged scientifically suitable for publication and will be formally accepted for publication once it meets all outstanding technical requirements.

Kind regards,

Vincenzo De Luca

Academic Editor

PLOS ONE

Additional Editor Comments (optional):

Reviewers' comments:

Reviewer's Responses to Questions

**Comments to the Author**

1. If the authors have adequately addressed your comments raised in a previous round of review and you feel that this manuscript is now acceptable for publication, you may indicate that here to bypass the “Comments to the Author” section, enter your conflict of interest statement in the “Confidential to Editor” section, and submit your "Accept" recommendation.

Reviewer #1: All comments have been addressed

2. Is the manuscript technically sound, and do the data support the conclusions?

Reviewer #1: (No Response)

3. Has the statistical analysis been performed appropriately and rigorously? 

Reviewer #1: (No Response)

4. Have the authors made all data underlying the findings in their manuscript fully available?

Reviewer #1: (No Response)

5. Is the manuscript presented in an intelligible fashion and written in standard English?

Reviewer #1: (No Response)

6. Review Comments to the Author

Reviewer #1: (No Response)

7. PLOS authors have the option to publish the peer review history of their article (what does this mean?). If published, this will include your full peer review and any attached files.

Reviewer #1: No

---

## [Editor Report · Acceptance letter]

30 Dec 2020

PONE-D-20-34056R1 

A systematic review and meta-analysis of victimisation and mental health prevalence among LGBTQ+ young people with experiences of self-harm and suicide. 

Dear Dr. Williams:

I'm pleased to inform you that your manuscript has been deemed suitable for publication in PLOS ONE. Congratulations! Your manuscript is now with our production department. 

Kind regards, 

on behalf of

Dr. Vincenzo De Luca 

Academic Editor

PLOS ONE